# Polylactide-Based Nonisocyanate Polyurethanes: Preparation, Properties Evaluation and Structure Analysis

**DOI:** 10.3390/polym16020253

**Published:** 2024-01-16

**Authors:** Anita Białkowska, Wojciech Kucharczyk, Iwona Zarzyka, Barbora Hanulikova, Milan Masař, Mohamed Bakar

**Affiliations:** 1Casimir Pulaski Radom University, 29 Malczewskiego Str., 26-610 Radom, Poland; wojciech.kucharczyk@uthrad.pl; 2Ignacy Łukasiewicz University of Technology in Rzeszow, 12 Powstańców Warszawy Str., 35-959 Rzeszów, Poland; izarzyka@prz.edu.pl; 3Tomas Bata University, Tr. Tomáše Bati 5678, 760 01 Zlín, Czech Republic; hanulikova@utb.cz (B.H.); masar@utb.cz (M.M.); 4Independent Researcher, 62-006 Gruszczyn, Poland

**Keywords:** nonisocyanate polyurethanes, polylactide, mechanical properties, thermal properties structure

## Abstract

This study investigated the successful synthesis and characterization of nonisocyanate polyurethanes (NIPUs) based on polylactide. The NIPUs were synthesized by a condensation reaction of oligomers with hard segments (HSs) and synthesized carbamate-modified polylactic acid containing flexible segments (FSs). The oligomers with HSs were prepared from phenolsulfonic acid (PSA) or a mixture of PSA and hydroxynaphthalenesulfonic acid (HNSA), urea and formaldehyde. The mixing of oligomeric compounds with different amounts of formaldehyde was carried out at room temperature. Obtained NIPU samples with different hard segment content were tested for their mechanical and thermal properties. The tensile strength (TS) of all NIPU samples increased with an increasing amount of HSs, attaining the maximum value at an HS:FS ratio of 1:3. Samples prepared from PSA and HNSA showed higher tensile strength (TS) without significant change in elongation at break compared to the samples based only on PSA. Thermogravimetric analysis data indicated an absence of weight loss for all samples below 100 °C, which can be considered a safe temperature for using NIPU materials. Maximum degradation temperatures reached up to 385 °C. Fourier transform infrared spectroscopy results confirmed the existence of expected specific groups as well as the chemical structure of the prepared polyurethanes. DSC analysis showed the existence of two characteristic phase transitions attributed to the melting and crystallization of hard segments in the NIPU samples.

## 1. Introduction

Nowadays, the challenge for researchers is to replace petroleum-based polymer components with renewable natural sources. Nonisocyanate polyurethanes (NIPUs) have attracted much attention over the past few decades due to growing environmental concerns. Some interesting reviews and research studies dedicated to the investigation of NIPUs have been published recently [1,2,3,4,5,6,7,8,9,10,11,12]. One of the most promising routes for the synthesis of NIPUs is the reaction between cyclic carbonates and amines without using toxic isocyanates. Endo et al. [13,14,15,16] have used various methods for the synthesis of isocyanate-free polyurethanes based on cyclic carbonates. They synthesized NIPUs by reacting five-membered cyclic carbonates with n-hexylamine [13]. Following this particularly interesting route with no hazardous isocyanates, polyhydroxyurethanes (PHUs) were obtained with inter- and intramolecular hydrogen bonds, which are expected to present higher chemical and hydrolysis resistances. In another study, Ochiai et al. [14] successfully synthesized PHUs using a polyaddition reaction of carbon dioxide, diamine and commercial bisepoxide. This method is one of the pioneering methods for the preparation of polyurethanes without using toxic and unstable diisocyanates. Various studies confirmed that the reaction conditions and components used influence the structure of the synthesized urethanes. Tomita et al. [15] studied the effect of reaction conditions on the structure of prepared polyhydroxyurethane and found that the structure was dependent on the solvent polarity and the amine structure. Moreover, it was shown that lower amounts of PHUs containing secondary hydroxyl groups were obtained when benzylamine was used instead of hexylamine. In a separate study, Tomita et al. [16] compared the reactions of six- and five-membered cyclic carbonates with hexylamine and benzylamine at different temperatures. Results showed that the six-membered cyclic carbonates were more reactive than the five-membered homologues and showed higher reaction rate constants. Moreover, the activation energy of the reaction of the six-membered ring carbonate with hexylamine was lower than that of the five-membered carbonate, due to the presence of the large ring strain of the former. Kim et al. [17] conducted a similar study and found that PHUs synthesized from aromatic diamines exhibited higher thermal stability than those from aliphatic amines due to the stiffness of the aromatic chain but with lower molecular weight. Lambeth and Henderson [18] investigated the effects of various catalysts on the reaction rates between cyclic carbonates and amines. Polymers synthesized at room temperature in the presence of triazabicyclodecene exhibited much higher molecular weight than those prepared without catalyst at both room temperature and 80 °C.

In recent years, vegetable oils, which are considered one of the cheapest and most widely available natural resources with biodegradability and limited toxicity, have been widely employed for the preparation of nonisocyanate polyurethanes (NIPUs). Studies using renewable resources for the synthesis of NIPUs have been carried out to meet the prerequisites of green chemistry and the needs of biodegradability of polymers [19,20,21,22]. Javni et al. [19] prepared NIPUs by reacting carbonated soybean oil with different diamines and studied the effect of amine structure and carbonate to amine ratio on NIPU mechanical and physical properties. All amines produced elastomeric polyurethanes with glass transitions between 0 °C and 40 °C and hardness between 40 and 90 Shore A. The same research group synthesized NIPUs by reacting carbonated soybean oil (CSBO) with aromatic and cycloaliphatic diamines [20]. NIPUs prepared with aliphatic diamines exhibited low tensile strength and low hardness values compared to those synthesized with diamines with an aromatic or rigid cyclic structure. The highest tensile strength was obtained with p-xylylene diamine and the lowest with m-xylylene diamine due to the differences in hydrogen bonding. However, elastomeric polyurethanes having glass transitions between −6 °C and 26 °C were obtained with all amines used. Bähr and Mülhaupt [21] studied the mechanical and thermal properties of NIPUs prepared by using CSBO and linseed (CLSO) oils with different diamines. It was shown that the glass transition temperature of the CSBO/isophorone diamine system increased by 40 °C and the Young’s modulus increased three orders of magnitude, while the elongation at break was reduced. Tamami et al. [22] confirmed that the physical properties of NIPUs synthesized from CSBO depended on the type of amine used. Farhadian et al. [23] prepared poly (ester amide/urethane) with vegetable oil without catalyst and without rigid and aromatic rings. Prepared NIPUs exhibited low water absorption and high molecular weight as well as improved thermal stability up to 386 °C. Kim et al. [17] reported a similar thermal stability improvement (up to 388 °C) for NIPUs based on rigid aromatic parts in their structures. Doley and Dolui [24] studied the effect of the amine structure and CSFO/amine molar ratio on the mechanical and chemical properties of NIPUs. NIPUs based on isophorone diamine and ethylenediamine displayed good thermal, mechanical, chemical and anticorrosive properties, whereas diethyltriamine-based NIPU exhibited high strain at break. Mahendran et al. [25] prepared a new bio-based nonisocyanate urethane through the aminolysis reaction of five-membered cyclic carbonate from linseed oil (CLSO) and phenalkamine at different cure temperatures. The impact strength of the coating slightly increased with an increase in amine concentration. The CLSO/phenalkamine ratio of 1:0.75 cured at 100 °C showed the best coating properties in comparison with all other CLSO/phenalkamine mixtures. Polyhydroxyurethanes (PHUs) were prepared by Mokhtari et al. [26] by aminolysis of highly reactive cyclic carbonates with various aliphatic and aromatic diamines at low temperature without catalyst. Cyclic carbonates with different functionalities were synthesized using Jojoba and castor oils. The thermal stability of synthesized PHUs was influenced by the structure of the diamine and the number of cyclic carbonates, and their glass transition temperatures ranged between −46 °C and 20 °C. A similar study was conducted by Jalilian and Yeganeh [27] who reacted a synthesized CSBO with amines at different weight ratios, atmospheric pressure and using a catalyst. Tensile strength and strain at break were shown to depend on the type of amine and their weight ratios with no clear trend. However, Lee and Deng [28] used lignin and soybean oil to prepare NIPU elastomer. Urethane bridges were formed by reacting CSBO with a coupling agent before lignin was introduced to produce the NIPU. The increase in lignin content resulted in an increase in the tensile strength and rigidity of prepared NIPUs. Similar results were obtained by Tavares et al. [29], who used technical Kraft lignin (TKL) with castor oil (CO) or modified castor oil (MCO) to synthesize NIPUs with improved properties. Salanti et al. [30] prepared various PHUs with various properties ranging from flexible elastomer to a thermosetting resin, by varying cyclocarbonates)/amines and poly(ethylene glycol)/lignin ratios. A lignin-based polyhydroxyurethane (LPHU) was prepared for the first time by Zhao et al. [31] from six-membered cyclic carbonate and isocyanate-free, solvent-free and catalyst-free fatty diamine dimer. The obtained LPHUs exhibited superior mechanical strength, high thermal stability, excellent recyclability and shape memory. In addition to vegetable oils, biodegradable polymers such as polycaprolactone and polylactide or other biodegradable natural materials have been used to make polyurethanes biodegradable and environmentally friendly [32,33,34,35,36].

The objective of this study was to explore an eco-friendly route to prepare a nonisocyanate polyurethane using biodegradable polylactic acid in the flexible segment. Recently, the environmental aspect, including the use of biodegradable polymers, has been taken very seriously by researchers.

## 2. Materials and Methods

### 2.1. Materials

The following chemicals were used in the present study:Phenol sulfonic acid as a 65% water solution, purchased from Sigma Aldrich Chemie GmbH (Steinheim, Germany);Urea and ethyl urethane from POCh (Polish Chemical Reagents Co., Gliwice, Poland);Formalin with concentration in the range 34–37% from Nitric Acid Plant (Tarnów, Poland);Tetrabutoxicin, from Merck Schuchardt (München, Germany);2-Naftol, Merck Schuchardt OHD (Hohenbrum, Germany);Sulfuric acid (96 wt% concentrated), Lach-Ner s.r.o. (Neratovice, Czech Republic);Lactid acid (L) in form of 80% solution in water from PENTA Gebäudeservice GmbH (Berlin, Germany);Poly(ethylene oxide) trade name PEG (Mn = 1000 g·mol^−1^; CAS NO 25322-68-3);Tin (II) 2-ethylhexanoate (Sn(Oct_2_)) from Sigma Aldrich Chemie GmbH (Steinheim, Germany);Methanol, chloroform from mikroCHEM (Olomouc—Hodolany, Czech Republic);Potassium hydroxide from Sigma Aldrich Chemie GmbH (Steinheim, Germany);Acetone, pyridine, acetanhydride, toluene from PENTA Gebäudeservice GmbH (Berlin, Germany).

### 2.2. Synthesis of Oligomeric Compounds Containing Hard Segments

The synthesis of nonisocyanate polyurethanes was carried out by condensation reaction of oligomeric compounds containing hard segments (HSs) and those with flexible segments (FSs).

Oligomers containing the hard segments (HSs) were prepared from urea, formaldehyde and phenol sulfonic acid (PSA) with or without hydroxyl naphthalene sulfonic acid (HNSA). The syntheses of the two oligomers have already been described in detail in our previous studies [37,38]. First, an oligomeric compound (designated HS1) based on urea, formaldehyde and PSA was prepared with molar ratio 1:1:2. Next, an oligomeric compound (designated HS2) based on a mixture of PSA and HNSA was prepared with the molar ratio of urea, HNSA, PSA and formaldehyde of 1:0.3:0.7:2. The reaction leading to oligomeric compounds HS1 and HS2 is shown in Figure 1.

### 2.3. Synthesis of Oligomeric Compounds Containing Flexible Segments

The flexible segment (FS) compounds were prepared from a derivative of poly (lactic acid)-PLA as follows: First, a poly(lactic acid)–poly(ethylene oxide) (PLA-PEG) copolymer (designated d-PLA) having reactive hydroxyl groups was synthesized, followed by amidation of hydroxyl groups to obtain urethane groups, capable of further reacting with oligomeric compounds containing hard segments.

The synthesis of poly(lactic acid)–poly(ethylene oxide) copolymer (d-PLA) was carried out as follows: 100 mL of lactic acid (L-LA) was introduced to a distillation flask equipped with a teflon stirrer which was fitted with a condenser and placed in an oil bath. First, the L-LA solution was dehydrated at 160 °C under a reduced pressure of 200 mbar for 3 h. Then, 0.5 mL of tin diacetate (Sn(Oct_2_)) was added as a catalyst and the reaction continued for 20 h under vacuum. In the next stage, the pressure was reduced to 0.1 mbar and a white powder was obtained [35].The synthesis of poly(lactic acid)–poly(ethylene oxide) copolymer (designated d-PLA) was carried out as follows: 100 mL of lactic acid (L-LA) was introduced to a distillation flask equipped with a teflon stirrer which was fitted with a condenser and placed in an oil bath. First, the L-LA solution was dehydrated at 160 °C under a reduced pressure of 200 mbar for 3 h. Then, 0.5 mL of tin diacetate (Sn(Oct_2_)) as a catalyst and 2 mol% of PEG were added and the reaction was continued for 20 h under vacuum. In the next stage, the pressure was reduced to 0.1 mbar and a white powder was obtained [35].After the polycondensation process, the d-PLA copolymer was subjected to purification. The copolymer was dissolved in acetone and introduced in a centrifuge using a 50% methanol solution in water for precipitation. Then, the obtained mixture was centrifuged 8 times in a centrifuge with 13,000 rpm for 5 min and at 25 °C, adding at regular intervals a new portion of the solvent (50% methanol solution in water). The same procedure was repeated using only water as the solvent. The sediment formed after centrifugation was spread on a large glass dish and dried for 48 h at 60 °C [27]. The obtained white powder was characterized for its hydroxyl number (LOH) and the molecular weight distribution.Amidation of hydroxyl groups of d-PLA containing flexible segments was obtained by reacting it with ethyl urethane according to Figure 2. A total of 1.4 moles of ethyl urethane was used per 1 mole of D d-PLA hydroxyl group in the presence of tin(IV) butoxide as a catalyst. The amidation of hydroxyl groups was carried out at the boiling point of the azeotrope using Tin (II) 2-ethylhexanoate (Sn(Oct_2_) as catalyst and toluene as solvent and azeotropic agent. The reaction was continued until the hydroxyl number was constant. The replacement of the hydroxyl groups with carbamate groups allowed the reaction of HS with the carbamate of the amidated compound (FS). Finally, the LOH of the obtained product was determined.

### 2.4. Preparation of Nonisocyanate Polyurethanes

The reaction between oligomeric compounds containing hard segments (HSs) with those having flexible segments (FSs) was possible only in the presence of reactive carbamine groups (Figure 3). The mixing of oligomeric compounds containing HSs and modified PLA carbamine with FSs and different amount of formaldehyde (stochiometric amount and 10% mole excess) was carried out at room temperature. The formed layers were left in a desiccator for 24 h and then heated in an air-circulating oven for different times (from 5 min to 180 min) and at different temperatures (from 25 °C to 80 °C) up to the solidification of samples, prior to mechanical testing and structure analysis.

### 2.5. Evaluation of Mechanical and Thermal Properties of NIPU Samples

Tensile tests were carried out using an Instron machine Model 5566 at room temperature and with tensile rate of 100 mm/min according to ISO/DIS 37 [39] standard. Tested NIPU films had 6cm in length, 1 cm in width and 0.3 mm in thickness.

Differential scanning calorimetry (DSC) was carried out with a Mettler Toledo DSC1 STAR testing device in the temperature range from −20 °C to 180 °C at a heating/ cooling rate of 10 °C/min and a nitrogen flow of 30 cm^3^/min. Initially, the NIPU samples were heated at 180 °C for 10 min, then cooled from 180 to −20 °C. The isothermal stage took place at −20 °C for a period of 10 min. The samples were then re-heated to 180 °C. The melting point and the enthalpy of fusion were read from the first heating cycle, while the glass transition temperature was determined from the second heating scan. The DSC analysis of the obtained polylactide films was performed under nitrogen conditions and gas stream of 60 mL/min. The samples were cooled from 25 to −60 °C, then heated to 25 °C, at a rate of 10 °C/min.

Thermogravimetric analysis (TG) curves of prepared NIPU samples were measured with a LabSys Evo DTA/DSC (Setaram, Sophia Antipolis, France). The tests were carried out under nitrogen atmosphere in temperature range 25–800 °C and heating rate of 10 °C/min.

The average molecular weight and molecular weight distribution were determined using a chromatograph-equipped Agilent GPC PL-GPC220 (Santa Clara, CA, USA) with an HT-GPC 220 detection system. The samples were dissolved in tetrahydrofuran (THF) at a concentration of ~3 g L^–1^ and separated on PL columns with mixed gel at 40 °C in THF at a flow rate of 1.0 mL /min.

### 2.6. Structure Characterization

Fourier transform infrared spectroscopy (FTIR) was performed using a Nicolet iS5 spectrophotometer (Thermo Fisher Scentific, Waltham, MA, USA) equipped with a germanium crystal in the wavenumber range 4000–700 cm^−1^. All spectrograms are the result of 64 scans. The tests were used to determine the chemical structure of d-PLA as well as NIPU samples.

## 3. Results and Discussion 

### 3.1. Structure Analysis of Synthesized Derivative Poly(lactic Acid)

The FTIR spectrum of the synthesized derivative of poly(lactic acid) (referred to as d-PLA) is shown in Figure 1. All characteristic bands of d-PLA were identified. 

The copolymer is characterized by the presence of end hydroxyl groups, which appear at 3550–3230 cm^−1^ (stretching vibrations of O-H groups). Overlapping bands from the vibrations of poly (lactide) and poly (ethylene oxide) groups have often been observed. This is the case for C-H stretching vibrations at 2800–3000 cm^−1^ of the methyl group in PLA and methylene in PEG. We also observed absorption peaks at 1760 cm^−1^ which are associated with C=O stretching vibrations as a result of the vibration of PLA carbonyl ester groups. Peaks located at 1400–1500 cm^−1^ and related to the vibration of CH_2_ bands of PEG and CH_3_ band of PLA have been identified. The peaks resulting from the vibration of stretching of groups C-O-C in the copolymer were identified in the range from 1050 to 1200 cm^−1^. In addition, weak signals arising from the deformation vibrations of the O-H and C-H groups were noticed in the range below 900 cm^−1^.

The estimated hydroxyl number of the synthesized polycondensation product d-PLA was 204 mg KOH/g, which confirmed its generated formula. It has to be pointed out that the reactive hydroxyl groups are very important for further synthesis.

Figure 2 shows the GPC chromatogram of the prepared d-PLA. The presence of a peak indicates a low molecular weight distribution in the resulted polymer. The product had a weight average molecular weight (M_w_) of 8200 g/mol. Its number average molar weight (M_n_) was ca 4700 g/mol and corresponded to the degree of polymerization (P_n_) of 66. The polydispersity index (PI) was lower than 2 (PI = 1.7), which means that the molecular weight distribution was relatively narrow. The result confirmed the correctness of the chosen method for carrying out the polycondensation reaction.

Based on the DSC thermogram shown in Figure 3, it can be concluded that d-PLA had three characteristic transition temperatures: glass transition (T_g_), crystallization (T_c_) and melting (T_m_).

The presence of a glass transition (T_g_) at 50 °C indicates the presence of flexible zones in the amorphous d-PLA. The crystallization at T_c_ = 95 °C confirms the possibility of ordering the d-PLA chains above T_g_. A double peak associated with melting may indicate the presence of two distinct crystalline phases. Nevertheless, the value of T_m_ at 156 °C can be used as a suitable estimate without further phase structure analysis. These values are in agreement with thermal properties results from other studies [35].

The results of the GPC analysis and the value of the hydroxyl number of d-PLA before and after amidation process (denoted as PLA-U) summarized in Table 1 made it possible to verify the accuracy of the applied method. It was found that the amidation reaction product (d-PLA) has 85% of amidated hydroxyl groups, which is confirmed by the hydroxyl number of d-PLA of 28 mg KOH/g. As can be seen, the molecular weight of the amidation product increased (relative to the molecular weight of d-PLA) by the value of attached carbamate groups. However, this increase is at the limit of the analysis error and thus cannot be overestimated. On the other hand, the modification process did not change significantly the degree of dispersity, confirming the lack of degradation of the d-PLA backbone.

### 3.2. Mechanical Properties of Segmented Nonisocyanate Polyurethanes

Figure 4 and Figure 5 show the effect of hard segment (HS) content on the tensile strength and strain at break of polyurethanes, respectively. It can be observed that the strength showed an upward trend while the strain at break presented a downward trend with increasing amounts of HS. Furthermore, the use of 10% molar excess of formaldehyde resulted in an increase in tensile resistance and reduction in relative elongation at break of the polymer, regardless of the type of HS used. In addition, all NIPU samples containing the mixture of phenolsulfonic acid (PSA) and hydroxyl naphthalene sulfonic acid (HNSA) (designated HS2) exhibited enhanced tensile strength and lower strain at break as compared to the polymers based on PSA without HNSA (HS1). Indeed, the strength of all NIPU samples containing three moles of HS increased by more than 100% in comparison with those samples with one mole of HS. The improvement in the TS and the reduction in the strain at break can be attributed to the crosslinking of the samples. Jin et al. [40] reported similar results for the tensile strength of conventional polyaddition thermoplastic polyurethane. They attributed the increase in tensile strength to the increase in hydrogen bonding between hard segments and the decrease in strain at break to the reduction in the rotation of segments in the chain.

### 3.3. Structure and Morphology Analyses of NIPU Films

Figure 6 shows FTIR spectra of nonisocyanate polyurethane (NIPU) samples containing a stoichiometric amount and 10% molar excess of formaldehyde designated “1” and “2”, respectively. Oligomers containing hard segments based on phenolsulfonic acid without hydroxyl naphthalene sulfonic acid (HNSA) were designated “I” (Figure 6a), while those containing HNSA were designated “II” (Figure 6b). Samples solidified for 24 h at room temperature and those subjected to heating at 50 °C for 2 h at 50 °C were designated “0” and “T”, respectively. It can be seen that the spectra and band intensities of the analyzed samples were similar without significant differences, regardless of the composition and preparation conditions.

The characteristic peaks recorded in the spectrograms appeared in the same wavenumber ranges, but only a few differed in intensity. The spectra of all NIPUs exhibited spectral bands in the regions 3000–2700 cm^−1^ and 1500–1350 cm^−1^ with bands of -CH_2_- stretching (2939 cm^−1^ and 2852 cm^−1^) and scissoring deformation bands. This is in accordance with the suggested structure of prepared NIPUs where the methylene bridges, the phenyl and/or hydroxynaphthalene and urethane groups, together with -CH_2_-CH_2_- units of the PEG part of FSs, were present. The contributions of C=O vibrations from both the PLA and urethane groups probably merged into a wide spectral band of the carbonyl group stretching at 1713 cm^−1^. In addition, the presence of the SO_3_H groups could be confirmed from the spectral bands in the region of 1300–900 cm^−1^. Although this region is also typical for C-O-C stretching of esters and urethanes, it is assumed that the very high intensity (and also the significantly lower intensity of C=O stretching band) rather indicates the contribution of OSO stretching and C-O-C stretching spectral bands overlap. Furthermore, the exposure of the samples to the increased temperature of 50 °C caused an increase in the intensity of the broad spectral band at 3600–2600 cm^−1^, indicating the presence of OH stretching of sulphonic acids. The OSO symmetric and asymmetric stretching region also showed a slight increase in intensity. Spectra of the samples, especially with segments HS2 (5-hydroxynaphthalene-2-sulphonic acid), provided the out-of-plane bending spectral bands of C-H at 850–550 cm^−1^ assigned to their aromatic parts.

FTIR analysis confirmed the expected structure of new PLA-based ionomeric condensation polyurethanes.

The results of thermogravimetric analysis are presented in Figure 7. As can be seen, major differences are evident between the two tested groups of samples. Values for weight loss and temperatures at which degradation occurred with maximum rate (T_d_) are presented in Table 2. The NIPU samples which contained hard segments based on urea and phenol sulfonic acid (I-1-0 and I-1-T) presented two distinct temperatures at around 270 °C and 385 °C, designated T_d2_ and T_d3_. Other gradual weight decreases (in Figure 7 on derivative weight seen as shoulders) at 120 °C–220 °C (T_d1_) precede these main weight losses. Moreover, it can be noted that higher amounts of formaldehyde in NIPUs (samples I-2-0 and I-2-T) caused a shift of the temperature T_d2_ to lower values of about 10 °C in comparison with samples with a stoichiometric amount of formaldehyde (samples I-1-0 and I-1-T).

However, NIPUs with hard segments based on urea, phenol sulfonic acid, formaldehyde and naphthalene sulfonic acid exhibited three relatively separated weight losses as a result of temperature increase (i.e., T_d1_ around 170 °C for samples designated II-1-0 and II-1-T and at 180 °C for samples II-2-0 and II-2-T). The second degradation degree T_d2_ was identified at 272 °C (for samples II-1-0 and II-1-T) and 265 °C (samples II-2-0 and II-2-T). The third step occurred at almost the same temperature for all samples and T_d3_ was at 385 °C. A temperature increase during the NIPU polycondensation (samples 0 and T) did not influence the evolution of the TGA curves. The main differences were evident between the tested samples I and II, and further in detail, between sets I-1 and I-2, and II-1 and II-2. On the other hand, almost no weight loss was observed in any samples below 100 °C. The relatively low thermal stability of NIPU can be attributed to the presence of a soft PEG segment in the PLA-PEG copolymer [41]. Puthumana et al. [42] reviewed the different copolymerization methods that have been used to overcome the limitations of PLA and thus expand its applications.

Figure 8 depicts DSC curves from heating and cooling of all studied NIPU samples. The values of melting (T_m_) and crystallization temperatures (T_c_) of samples are included in Table 3. We can see that there are two characteristic phase transitions: melting (T_m_) and crystallization (T_c_). The crystallization process took place in a narrow range from −2 to +2 °C and melting in the range of +19 to +21 °C.

It can be observed that the NIPU composition and the preparation conditions had a slight influence on the T_c_ values (Table 3). The NIPU obtained from only phenolsulfonic acid (HS1) showed lower T_c_ values than that obtained from HS2 (based on naphthalenesulfonic acid and phenolsulfonic acid). Moreover, heating the samples at 50 °C for 2 h increased slightly the T_c_ in NIPU obtained from a stoichiometric amount of formaldehyde, while in samples with a higher degree of cross-linking, it did not affect the crystallization temperature. On the other hand, there was no noticeable influence of the composition and heating on the recorded melting. However, the presence of one characteristic crystallization temperature and one melting temperature indicates the non-segmented structure of the obtained PLA-based polyurethanes. The analysis of the results suggested that the phase transformations of melting and crystallization phenomena as well as the corresponding T_m_ and T_c_ can be attributed to phase transformations of the PLA copolymer used for the synthesis.

However, the above transformations were not identified for the HSs incorporated into the obtained polymer. It may be related to the lack of phase separation in the tested polymer, which is characteristic of analogous NIPUs [37,38].

It has been reported that to increase the melting temperature and ductility of semi-crystalline PLA, other comonomers such as trimethylene carbonate should be used [43].

## 4. Conclusions

The present study confirmed the successful synthesis of poly (lactic acid) copolymer (d-PLA) as a base constituent for segmented nonisocyanate polyurethanes (NIPUs). The molecular weight of d-PLA was estimated and its structure confirmed. NIPUs were synthesized by the condensation reaction of oligomers containing hard segments with those having flexible segments. The former were prepared from urea, formaldehyde and either phenol sulfonic acid (PSA) or a mixture of PSA and hydroxyl naphthalene sulfonic acid (HNSA). However, the flexible segments were obtained from modified PLA. All NIPU samples containing both PSA and HNSA exhibited improved tensile strength (TS) as compared to the polymers based on PSA without HNSA. A wide application temperature range for the prepared polyurethanes was defined by a glass transition at 50.7 °C and a melting temperature up to 156 °C. Very high degradation temperatures were noted for NIPU samples based on PSA and HNSA, specifying the high thermal stability of the materials tested. Phenolsulfonic acid (PSA)-based NIPU samples did not exhibit a different morphology from that obtained from a mixture of PSA and hydroxynaphthalene sulfonic acid (HNSA).

This study confirmed that it is possible to prepare environmentally friendly polyurethanes having interesting performance properties by using a biodegradable polymer and a nonisocyanate route.

## Data Availability

Data available on request.

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
