# Peer review of "Polylactide-Based Nonisocyanate Polyurethanes: Preparation, Properties Evaluation and Structure Analysis"

_polymers, 2024, doi:10.3390/polym16020253_

Round 1
Reviewer 1 Report
Comments and Suggestions for Authors
This paper should be published. The only issue is the grammar and spelling mistakes throughout this manuscript. A review of the English grammar/spelling should be conducted by someone proficient in the English language
Comments on the Quality of English LanguageA review of this manuscript should be completed by the authors regarding use of the English language.
Author Response
Thank very much for your time and recommendation which was very helpful in improving our manuscript. The manuscript has been double-checked and we have made all necessary corrections.

Reviewer 2 Report
Comments and Suggestions for Authors
This paper presented a PLA-based non-isocyanate polyurethane.
I don’t think this work has any meaningful scientific rationale and merits. Please allow me question the authors by these equestions/comments.
(1) It is true that NIPU is considered “green” compared to conventional isocyanate resins. However, in this study, I don’t see any advantage. It uses more toxic chemicals (especially formaldehyde).
(2) Again, although the authors don’t give detailed quantities, the amount of PLA in the overall NIPU is very low. I also don’t understand why it’s a poly (lactic acid)-poly (ethylene oxide) copolymer and not pure PLA. Perhaps someone has already developed a 100% PLA-based NIPU? Also, D-PLA is white powder at room temperature, then how to mix it as a traditional polyol?
(3) No analytical characterization of the synthetic chemicals (including “HS”, “D-PLA” and NIPU) has been done. Why there is no NMR results? And GPC results of NIPU (molecular weight)?
(4) It is uncertain whether this PLA-based NIPU is cross-linked or linear, especially given its synthetic route.
(5) From the DSC results, this PLA-based NIPU is a linear polymer. However, its low melting point and crystallization temperature severely limit its application. I don’t know what use a polymer with a melting point of only 25 °C can have. Also, why NIPUs have an even lower melting point than PLA oligomer (as “soft segment”)?
(6) Similarly, the thermal stability (expressed as TGA) of NIPU is extremely poor. I don’t see any potential use for it. Even as an adhesive, degradation at about 100 °C makes it difficult to use. Likewise, significant loss at ~120 °C (supposed to be under nitrogen testing environment) suggests that such NIPU may degrade at even ~80 °C in reality.
(7) The overall synthesis route is way so complicated and impractical. What is the point of introducing sulfonic groups in polyurethane? What is the final product of sulfonic acid after the reaction (sulfonic acid is highly reactive). The synthesis of oligomeric compounds with HS is also incorrect. Same for the synthesis of D-PLA by ethyl urethane, why the NH2 end group will not react with the -OH end group instead of transesterification? There are no analytical results!
(8) SEM images did not indicate anything.
The final NIPU material has no use in any application due to its extremely poor thermal properties (uncertain about whether the polymerization is complete). I also question the synthesis methods of NIPUs. But, in general, using formaldehyde to make “non-isocyanate” polyurethane has no practicability or sustainability feature, therefore I don’t see any potential.
Comments on the Quality of English LanguageThe English writing in this paper is decent. However, minor improvement is suggested.
Author Response
Thank You very much for your recommendations and remarks. We have responded point by point to all the suggestions. Our responses are in blue below, while changes and added explanations are also in blue in the manuscript.

Reviewer 3 Report
Comments and Suggestions for Authors
In general, the work is quite good, but there are a number of unclear points.
1) If you are claiming that a given NIPU is biodegradable, wouldn't it be nice to show studies on its biodegradation? or will you present the data in a subsequent paper?
2) Why did you use phenol sulfonic acid and hydroxyl naphthalene sulfonic acid as reagents?
3) The resulting NIPU contains sulfur in its composition. After decomposition, for example in the environment, how will the decomposition products affect soil or water and will it be environmentally friendly? But if I understand correctly, environmentally friendly polyurethane in your case it is polyurethane without the use of traditional isocyanate.
4) Maybe I missed something, but in section 2.3. you describe the preparation of a copolymer. But in this section there is nothing about obtaining a copolymer. No mention of the introduction of ethylene oxide.
5) You name the copolymer as D-PLA. But traditionally this is called polylactide from D lactic acid or D lactide. But in the first part of 2.3. (2.3.1) you indicate the use of L lactic acid.
6) Discussion of the FTIR results in section 3.1. not entirely clear. There is also no clear confirmation of the structure and composition of the copolymer. What is the ratio of lactic acid and ethylene glycol groups in the copolymer? It might be a good idea to test the sample using NMR.
7) The reaction conditions for the amidation of hydroxyl groups are not described in 3.2.3. Lines 193-194 ...The amidation of hydroxyl groups was carried out at the boiling point of the azeotrope. .... The boiling point of the azeotrope with what substance?
Author Response
Thank very much for your recommendations and remarks which was very helpful in improving our manuscript. We have responded point by point to all the recommendations and remarks. Our responses to the reviewers are in blue below, while changes and added explanations are also in blue in the manuscript
